# Who gets missed in Zambia's antenatal HIV testing? Persistent inequities despite routine screening policies in Zambia

Newton Nyirenda[1]*, Whiteson Mbele[2], Azielyn Mutiibwa[3], Hannah Muturi[4], Linda Siachalinga[5]

1 Department of Minority Health and Health Disparities Research, Georgetown University, Washington, District of Columbia, United States of America, 2 Ministry of Health Zambia, Kasiya Mini Hospital, Pemba, Zambia, 3 University of Northern Philipines, Tagum, Philipines, 4 Department of Science and Mathematics, University of the District of Columbia, Washington, District of Columbia, United States of America, 5 School of Medicine and Dentistry, Griffith University, Gold Coast Campus, Queensland, Australia

* nan53@georgetown.edu

## Abstract

Although substantial advances have been made in eliminating mother-to-child HIV transmission in sub-Saharan Africa, significant disparities in antenatal HIV testing by socioeconomic status and geographic location continue. We examined predictors of HIV testing during pregnancy among women in Zambia, with a focus on equity-related factors, utilizing data from the 2007, 2013–2014, and 2018 rounds of the Zambia Demographic and Health Surveys, which are nationally representative household surveys. Eligible participants were women aged 15–49 who reported a live birth in the five years preceding the survey and received at least one antenatal care (ANC) visit during that pregnancy. We applied survey-adjusted logistic regression models to analyze time trends and identify key sociodemographic factors associated with HIV testing uptake. We also conducted stratified analyses by urban–rural residence. The study sample consisted of 4047 participants in 2007, 9190 in 2013/2014 and 7245 in 2018. Self-reported HIV testing among women who attended antenatal care and had recent births rose from 87% in 2007 to 95% in 2018. However, persistent disparities were observed. Women with no education, in the poorest wealth quintile, or residing in rural areas were significantly less likely to be tested. In multivariable models, education and wealth were strong predictors of testing uptake. Stratified models revealed that education and wealth gradients were steeper in rural than urban areas. While Zambia has made major gains in antenatal HIV testing coverage, persistent inequities remain among the poorest, least educated, and rural-dwelling women. To close these gaps, national policy should prioritize community-based testing integrated within ANC services, expand the use of mobile clinics in rural areas, and implement peer-led education initiatives, approaches shown in other sub-Saharan African

**Data availability statement:** The datasets used in this study are publicly available via the DHS Program website: https://dhsprogram.com/data/available-datasets.cfm.

**Funding:** The authors received no specific funding for this work.

**Competing interests:** The authors declare that they have no competing interests.

settings to improve testing uptake among underserved women. These equity-focused strategies are essential to achieving universal HIV testing in pregnancy and advancing Zambia's PMTCT and 95–95–95 goals.

## Introduction

Human Immune Deficiency Virus (HIV) screening as part of antenatal care (ANC) is an essential public health strategy for preventing mother-to-child transmission (PMTCT) of HIV. Early identification of maternal HIV status enables timely initiation of antiretroviral therapy (ART), reducing perinatal transmission risk to less than 1% with optimal care [1,2]. As a result, universal HIV screening ANChas been endorsed by the World Health Organization (WHO) as part of routine maternal health services [1]. In sub-Saharan Africa, where over 90% of the world's paediatric HIV infections occur, integrating HIV testing into ANC remains vital to controlling the epidemic [3].

Zambia has made notable progress in expanding ANC-based HIV testing as part of its PMTCT strategy. According to national guidelines, all pregnant women should be tested for HIV at their first ANC visit, with re-testing offered later in pregnancy [4]. Consequently, HIV testing coverage among pregnant women increased dramatically between 2007 and 2018, aided by the scale-up of provider-initiated testing and counselling (PITC), community mobilization, and ART decentralization [5]. However, progress has not been uniform, and emerging evidence suggests persistent inequities in access to maternal HIV services based on socioeconomic and geographic factors. Barriers to testing include inadequate health literacy, fear of stigma or disclosure, gendered power imbalances such as male partner influence on ANC decisions, and structural constraints like stock-outs of test kits or long travel distances to facilities [6].

Prior studies from Zambia and other low- and middle-income countries (LMICs) have documented disparities in HIV testing uptake by educational attainment, household wealth, and urban–rural residence [7–9]. Women with lower education or income levels may face greater barriers to testing, including limited health literacy, structural discrimination, stigma, and logistical challenges such as transport and service availability [10–12]. Even among ANC attendees, these factors may lead to missed testing opportunities and undermine the effectiveness of PMTCT programs [13,14].

Despite growing interest in addressing maternal health inequities, there is limited recent evidence on how the equity landscape of antenatal HIV testing has evolved over time in Zambia. Most demographic health surveys (DHS)-based studies have either focused on single cross-sections or broader determinants of HIV testing without stratifying by pregnancy status or ANC attendance [15]. Understanding how patterns of testing among ANC users have shifted over time, and whether key equity gaps persist, can provide actionable insights for policy and implementation.

This study addresses that gap by using three rounds of nationally representative DHS data from Zambia (2007, 2013–14, and 2018) to (1) assess trends in HIV testing coverage among women with a recent birth who attended ANC, and (2) identify sociodemographic factors associated with HIV testing uptake over time. We further stratify the analysis by place of residence to examine whether observed disparities

differ between rural and urban settings. Our findings aim to inform equity-oriented strategies for achieving universal HIV testing in pregnancy, aligned with the goals of the United Nations Programme on HIV/AIDS (UNAIDS) 95–95–95 targets and the Zambia National Human Immunodeficiency Virus/Acquired Immunodeficiency Syndrome Strategic Framework.

## Methods

### Ethics statement

This analysis utilized anonymized secondary data from the publicly available DHS Program database. Datasets were obtained from the DHS Program website on 25th September 2024, following registration and data access approval. Ethical clearance for DHS data collection was obtained from the DHS Program and from the University of Zambia Biomedical Research Ethics Committee and the Tropical Diseases Research Centre Ethics Review Committee, which were responsible for reviewing protocols at the time of each survey. No additional ethical approval was required for this secondary analysis.

### Study design

This was a pooled cross-sectional study using data from the 2007, 2013–2014, and 2018 Zambia DHS The DHS is a nationally representative household survey that collects information on health, fertility, maternal and child health, and HIV indicators. It uses a two-stage stratified sampling design and includes data weighted to account for complex survey design, clustering, and oversampling.

This study focused on 20,482 women aged 15–49 who had a live birth in the five years preceding the survey and reported attending at least one ANC visit. These inclusion criteria were used to ensure the population had reasonable opportunity to be offered HIV testing during pregnancy. For each survey, we extracted the women's recode files (IR files) and harmonized variables across survey years to ensure comparability.

### Outcome and explanatory variables

The main outcome was HIV testing during pregnancy, based on self-reported receipt of an HIV test and result during the most recent pregnancy. This variable was derived from standardized DHS items collected during ANC visits (e.g., m49a_1, recoded as binary).

Key explanatory variables included survey year (2007, 2013/14, 2018), age category (<20, 20–34, ≥ 35), level of education (no formal education, primary, secondary, higher), household wealth index (poorest to richest), and place of residence (urban vs rural). These were constructed using DHS standard variables and selected based on prior literature highlighting their relevance to equity in maternal health service access, including studies demonstrating disparities in HIV testing uptake by age, education, wealth, and residence [7,10,16,17].

### Statistical analysis

We adjusted for the complex sampling structure of the DHS by applying survey weights and incorporating design elements for clustering and stratification (v005, v021, and v022). Descriptive analyses were conducted by survey year using weighted proportions to summarize sociodemographic characteristics. HIV testing prevalence and corresponding 95% confidence intervals were then estimated overall and by urban–rural residence. To identify predictors of HIV testing, we fitted multivariable logistic regression models using survey weights and quasi-binomial error structures, which accounted for modest overdispersion observed in preliminary models and provided more reliable standard errors.. Adjusted odds ratios (aORs), 95% confidence intervals (CIs), and p-values were reported. We assessed multicollinearity using variance inflation factors (VIF). Model diagnostics were assessed using McFadden's pseudo-$R^2$ and the area under the ROC curve (AUC).

Separate models were run for rural and urban populations to assess potential differences in associations by residence. In addition, interaction terms between education and residence, and wealth and residence, were tested to evaluate whether disparities differed significantly by geography.

Analyses were conducted in R version 4.3.1 using the survey, dplyr, and gtsummary, and pROC packages [18].

## Results

Table 1 summarizes weighted background characteristics of women aged 15–49 with recent births and ANC attendance, across the 2007–2018 DHS rounds. Across all survey years, the majority of women were aged 20–34 years, with a modest increase in the proportion of adolescent mothers (<20 years) from 8.3% in 2007 to 9.8% in 2018. Educational attainment improved over time, with the proportion of women reporting secondary or higher education rising from 26% in 2007 to 41% in 2018, while those with no formal education decreased from 13% to 9.1%. Wealth distribution remained relatively stable across survey rounds. The percentage of women residing in urban areas increased slightly, from 33% in 2007 to 39% in 2018. Notably, the proportion of women who reported being tested for HIV during pregnancy increased from 87% in 2007 to 95% in 2018, representing a substantial gain in coverage This improvement coincided with rising educational attainment, particularly in rural areas, suggesting that greater access to schooling may have contributed to higher testing uptake over time (Table 2) (Fig 1).

Multivariable logistic regression analysis indicated that HIV testing uptake during pregnancy increased significantly over time. Compared to 2007, the odds of being tested for HIV were more than twice as high in 2013 (OR = 2.3, 95% CI: 1.9–2.8) and three times as high in 2018 (OR = 3.0, 95% CI: 2.4–3.8). Maternal age was also associated with HIV testing: women aged 20–34 and those 35 years or older had significantly higher odds of testing compared to those under 20 years of age (OR = 1.5 and 1.4, respectively).

**Table 1. Sociodemographic characteristics of women with a recent birth and antenatal care attendance, Zambia DHS 2007–2018.**

| Characteristic | 2007 (N = 4,047) | 2013/14 (N = 9,190) | 2018 (N = 7,245) |
|---|---|---|---|
| **Age** | | | |
| <20 | 337 (8.3%) | 833 (9.1%) | 713 (9.8%) |
| 20–34 | 2,885 (71%) | 6,233 (68%) | 4,861 (67%) |
| 35+ | 825 (20%) | 2,125 (23%) | 1,671 (23%) |
| **Education** | | | |
| No education | 510 (13%) | 917 (10.0%) | 662 (9.1%) |
| Primary | 2,471 (61%) | 4,919 (54%) | 3,556 (49%) |
| Secondary | 940 (23%) | 2,976 (32%) | 2,711 (37%) |
| Higher | 126 (3.1%) | 368 (4.0%) | 315 (4.4%) |
| **Wealth Quintile** | | | |
| Poorest | 892 (22%) | 1,997 (22%) | 1,640 (23%) |
| Poorer | 836 (21%) | 1,932 (21%) | 1,509 (21%) |
| Middle | 818 (20%) | 1,903 (21%) | 1,379 (19%) |
| Richer | 847 (21%) | 1,775 (19%) | 1,459 (20%) |
| Richest | 654 (16%) | 1,583 (17%) | 1,258 (17%) |
| **Residence** | | | |
| Rural | 2,713 (67%) | 5,693 (62%) | 4,453 (61%) |
| Urban | 1,334 (33%) | 3,497 (38%) | 2,792 (39%) |
| **HIV Tested During Pregnancy** | | | |
| Yes | 3,507 (87%) | 8,623 (94%) | 6,908 (95%) |

**Table 2. Weighted multivariable logistic regression predicting HIV testing during pregnancy, Zambia DHS 2007–2018.**

| Characteristic | OR | 95% CI | p-value |
|---|---|---|---|
| **Survey Year** | | | |
| 2007 (Ref) | — | — | — |
| 2013 | 2.3 | 1.9, 2.8 | <0.001 |
| 2018 | 3.0 | 2.4, 3.8 | <0.001 |
| **Age Group** | | | |
| <20 (Ref) | — | — | — |
| 20–34 | 1.5 | 1.2, 1.8 | <0.001 |
| 35+ | 1.4 | 1.1, 1.9 | 0.008 |
| **Education Level** | | | |
| No education (Ref) | — | — | — |
| Primary | 1.6 | 1.3, 1.9 | <0.001 |
| Secondary | 2.2 | 1.8, 2.8 | <0.001 |
| Higher | 1.8 | 1.1, 3.0 | 0.020 |
| **Wealth Quintile** | | | |
| Poorest (Ref) | — | — | — |
| Poorer | 1.4 | 1.2, 1.7 | <0.001 |
| Middle | 1.7 | 1.4, 2.1 | <0.001 |
| Richer | 1.5 | 1.2, 1.9 | 0.001 |
| Richest | 1.4 | 0.99, 2.0 | 0.061 |
| **Place of Residence** | | | |
| Rural (Ref) | — | — | — |
| Urban | 1.1 | 0.91, 1.4 | 0.300 |

*OR = Odds Ratio; CI = Confidence Interval; Ref = Reference category.

Educational attainment showed a strong, positive gradient, with increasing levels of education associated with higher odds of HIV testing. For instance, women with secondary education had over twice the odds of being tested compared to those with no formal education (OR = 2.2, 95% CI: 1.8–2.8). Household wealth was positively linked with HIV testing uptake. Women belonging to the middle and higher wealth quintiles were significantly more likely to be tested during pregnancy than those from the lowest quintile. However, the association for the richest quintile was borderline significant (OR = 1.4, 95% CI: 0.99–2.0; $p = 0.061$). Place of residence was not significantly associated with HIV testing after adjusting for other factors (OR = 1.1, 95% CI: 0.91–1.4).

Model diagnostics indicated modest explanatory power (McFadden's pseudo-$R^2 = 0.045$). Discrimination was fair, with an AUC of 0.66. All VIF values were <2.0, suggesting no problematic multicollinearity (Fig 2). Stratified models revealed key differences in predictors of HIV testing during pregnancy between rural and urban areas (Table 3). In rural settings, higher education and greater household wealth were strongly associated with increased odds of testing. Women with secondary education had more than twice the odds of HIV testing compared to those with no education, and the effect of wealth was most pronounced in the richer quintiles. In contrast, these associations were attenuated and not statistically significant in urban areas. Education level was not a significant predictor of HIV testing among urban women, and only those in the poorer and middle wealth quintiles had elevated odds of testing compared to the poorest. Age-related differences were also more pronounced in rural areas, where older women were significantly more likely to be tested than adolescents, whereas no significant age effects were observed in urban areas. The trend

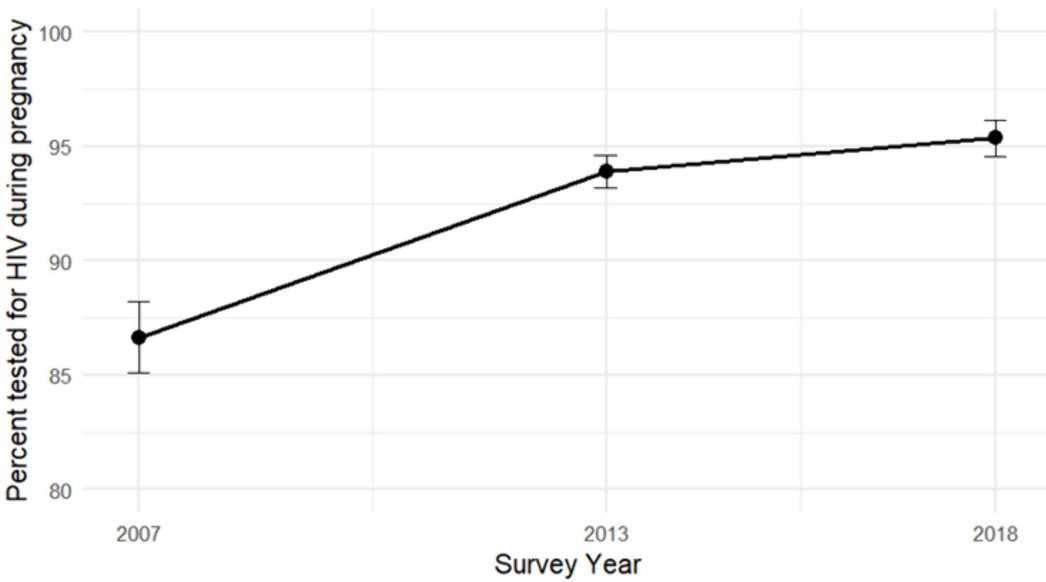

**Fig 1. Trends in HIV testing during pregnancy among women aged 15–49 with a live birth in the five years preceding the survey and at least one antenatal care visit.** Estimates are based on weighted analysis of the 2007, 2013/14, and 2018 Zambia Demographic and Health Surveys. Error bars represent 95% confidence intervals, adjusted for complex survey design.

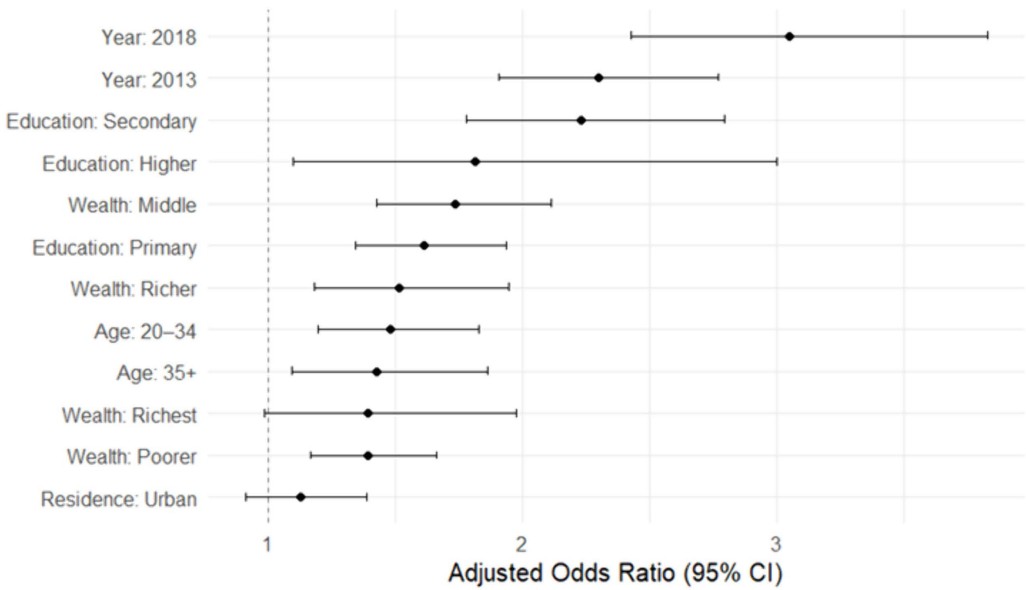

**Fig 2. Adjusted odds ratios and 95% confidence intervals for HIV testing during pregnancy among women with a recent birth who attended antenatal care.** Estimates are presented as a forest plot for improved clarity, based on a weighted multivariable logistic regression model using data from the 2007, 2013/14, and 2018 Zambia Demographic and Health Surveys. The model adjusts for survey year, age, education, wealth quintile, and place of residence. The vertical dashed line indicates the null value (OR = 1).

**Table 3. Stratified multivariable logistic regression predicting HIV testing during pregnancy by place of residence, Zambia DHS 2007–2018.**

| Characteristic | Rural Areas OR (95% CI) | p-value | Urban Areas OR (95% CI) | p-value |
|---|---|---|---|---|
| **Survey Year** | | | | |
| 2007 (Ref) | — | — | — | — |
| 2013 | 2.10 (1.69, 2.61) | <0.001 | 2.82 (1.94, 4.09) | <0.001 |
| 2018 | 3.64 (2.86, 4.64) | <0.001 | 2.14 (1.37, 3.32) | <0.001 |
| **Age Group** | | | | |
| <20 (Ref) | — | — | — | — |
| 20–34 | 1.75 (1.38, 2.21) | <0.001 | 0.93 (0.58, 1.50) | 0.80 |
| 35+ | 1.74 (1.31, 2.31) | <0.001 | 0.81 (0.44, 1.50) | 0.50 |
| **Education Level** | | | | |
| No education (Ref) | — | — | — | — |
| Primary | 1.72 (1.42, 2.09) | <0.001 | 0.87 (0.46, 1.63) | 0.70 |
| Secondary | 2.65 (2.02, 3.47) | <0.001 | 1.10 (0.59, 2.04) | 0.80 |
| Higher | 2.24 (0.73, 6.86) | 0.20 | 0.95 (0.43, 2.13) | >0.90 |
| **Wealth Quintile** | | | | |
| Poorest (Ref) | — | — | — | — |
| Poorer | 1.37 (1.14, 1.64) | <0.001 | 2.64 (1.09, 6.40) | 0.031 |
| Middle | 1.58 (1.30, 1.93) | <0.001 | 3.65 (1.51, 8.79) | 0.004 |
| Richer | 1.95 (1.43, 2.67) | <0.001 | 2.12 (0.88, 5.08) | 0.093 |
| Richest | 1.42 (0.65, 3.10) | 0.40 | 2.22 (0.90, 5.51) | 0.085 |

*OR = Odds Ratio; CI = Confidence Interval; Ref = Reference category.

of increased HIV testing over time was evident in both settings, but the magnitude of change was greater in rural areas, suggesting improved outreach or policy reach in underserved populations (Fig 3).

Interaction models testing education x residence and wealth x residence terms showed limited evidence of effect modification: the secondary education x urban interaction reached statistical significance (p=0.017), whereas most other interactions were not significant. For interpretability, we retained stratified models in the main text but also present predicted probability plots (Fig 4) to illustrate potential geographic differences. Full results from interaction models are provided in S1 Table.

## Discussion

The analysis of Zambia DHS data from 2007 to 2018 indicates a substantial improvement in HIV testing during pregnancy, increasing from approximately 87% to over 95%. This trend aligns with pan-African patterns, where many countries have achieved near-universal antenatal HIV screening, although uneven across regions [16,19]. The adoption of Option B+ in Zambia in 2012, providing lifelong ART to pregnant women, likely facilitated this advance by integrating HIV testing into antenatal services [20,21]. National PMTCT initiatives also appear to have bolstered testing uptake during this period [22].

Despite these gains, significant disparities in uptake of recommended testing persist even among women engaged in ANC, highlighting that access does not eliminate inequities in maternal health services. Our results demonstrate a robust education gradient: women with secondary or higher schooling had over twice the odds of testing compared to those with no formal education, echoing findings from other East African DHS-based studies [23,24]. Wealth-related disparities were also evident; middle and richer quintile women were more likely to test, consistent with broader regional analyses demonstrating pro-rich testing bias [19,23]. These findings highlights systematic and structural factors that need to be addressed, even among women attending ANC [25]. In one study, globally, 72.9% of women who attended ANC reported receiving essential services, but this was highly inequitable especially in LMICS [26]. This highlights the fact that increasing ANC coverage without addressing quality and equity gaps will not achieve the desired health coverage goals.

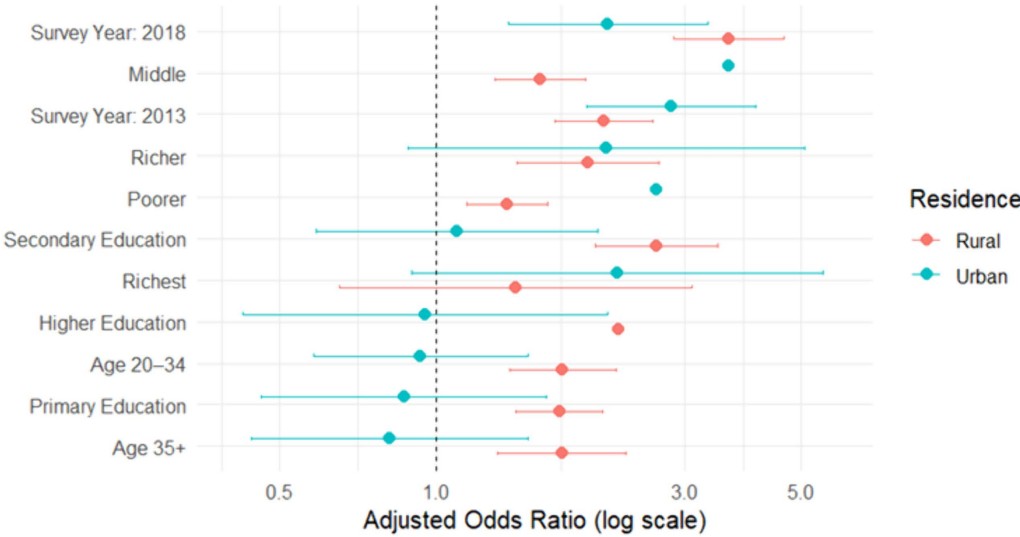

**Fig 3. Stratified adjusted odds ratios (aORs) for HIV testing during pregnancy by place of residence, Zambia DHS 2007–2018.** Separate multivariable logistic regression models were fit for rural and urban subpopulations, adjusting for survey year, age, education level, and household wealth quintile.

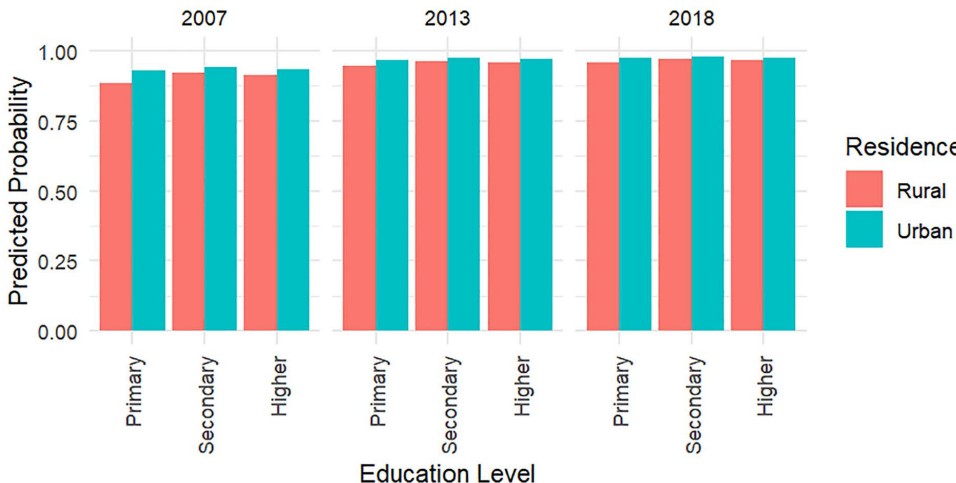

**Fig 4. Predicted probabilities of HIV testing during pregnancy by education level and place of residence across survey years, Zambia DHS 2007-2018.** Estimates are based on survey-weighted logistic regression models including interaction terms between education and residence.

Age was another critical determinant, with older women showing higher testing rates, especially in rural areas, suggesting outreach strategies may be missing adolescents. Similar concerns have been reported across sub-Saharan Africa, where youth access remains limited [19,27]. This could be due to reported challenges that adolescent may face including age-related stigma, low heath literacy, financial burden, limited autonomy and fear [28]. Peer-led, age-appropriate PMTC services can help to bridge communities and healthy facilities hence addressing systematic barriers to care. Alternatively, Peer Navigators could conduct extensive outreach at adolescent congregation points [29]. Increasing knowledge of PMTC

through community-based intervention delivered in community gatherings could increase the service uptake by adolescents [30].

### Rural-urban differences

Stratified analysis by residence revealed sharper socioeconomic disparities in rural than in urban areas. In urban settings, education and wealth had minimal predictive value, while in rural contexts both were strong predictors. This suggests that access barriers, due to infrastructure, information, or resources, are more pronounced outside urban centers. These findings mirror other studies illustrating how rural disadvantage amplifies socioeconomic gradients in health service uptake [16,19].

Our results have important equity implications: while overall testing coverage is high, gaps in rural, less educated, and financially disadvantaged women remain. To support equity, Zambia could strengthen its national strategy by expanding rural outreach through community health workers and mobile testing clinics. Peer-led education programs tailored for women with limited formal schooling may also improve awareness and acceptance of antenatal HIV testing. In addition, integrating community-based testing into other routine maternal services can reduce both stigma and logistical barriers to uptake. The USAID Global Health Supply Chain Program (GHSC-PSM) emphasizes the significance of a stronger public health supply chain as essential for achieving HIV testing and treatment targets [31,32], this could build trust in the reliability of the healthcare system increase compliance to follow-up.These approaches are aligned with WHO and United Nations Children's Fund (UNICEF) frameworks that emphasize the need for person-centered, context-specific care [1,33]. Furthermore, international evidence supports the effectiveness of community-driven, education-sensitive models in closing gaps in prenatal HIV testing [23,27].

This study has several limitations. A key limitation of this study is that HIV testing was self-reported, which may introduce recall inaccuracies and social desirability bias (SDB). SDB may vary by context and in some cases underestimate disparities across groups [34,35]. Second, the use of cross-sectional DHS data restricts our capacity to determine causal links between predictors and HIV testing outcomes. Third, we did not model geographic or regional clustering, which has been highlighted as a limitation in similar DHS-based national studies [2].

Despite these limitations, the study has notable strengths. It draws on three waves of nationally representative DHS data with consistent measures over time, allowing robust comparison of trends across more than a decade. The large sample size and use of appropriate survey weights enhance generalizability, and the stratified analysis by place of residence provides insight into urban–rural disparities that are often obscured in pooled estimates. Furthermore, by focusing on women with recent births and ANC contact, the analysis highlights equity gaps within a group already assessing ANC services, offering clear direction for targeted interventions.

### Conclusion

HIV testing during pregnancy in Zambia has expanded substantially from 2007 to 2018, reflecting national progress toward PMTCT goals. However, despite high overall coverage, persistent inequities remain, particularly among women lower socioeconomic status and residing in rural areas. These disparities are especially concerning given that all women in this analysis were already assessing ANC services. To achieve equitable maternal HIV prevention, future programs must go beyond expanding access and address the structural, educational, and geographic barriers that limit universal testing uptake. Integrating community-based, education-sensitive strategies within antenatal services can accelerate progress toward more inclusive maternal health outcomes.

### Supporting information

**S1 Table.  Multivariable logistic regression with interaction terms for HIV testing during pregnancy, Zambia DHS 2007–2018.**
(DOCX)

## Author contributions

**Conceptualization:** Newton Nyirenda, Whiteson Mbele.

**Formal analysis:** Newton Nyirenda.

**Resources:** Whiteson Mbele.

**Writing – original draft:** Newton Nyirenda.

**Writing – review & editing:** Azielyn Mutiibwa, Hannah Muturi, Linda Siachalinga.

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
