## [Decision Letter · Decision Letter 0]

21 Aug 2025

PGPH-D-25-01877

Who Gets Missed? Inequities in Antenatal HIV Testing in Zambia Despite Routine Screening Policies.

Dear Dr. Nyirenda,

Thank you for submitting your manuscript to PLOS Global Public Health. After careful consideration, we feel that it has merit but does not fully meet PLOS Global Public Health’s publication criteria as it currently stands. Therefore, we invite you to submit a revised version of the manuscript that addresses the points raised during the review process.

We look forward to receiving your revised manuscript.

Kind regards,

Tamara Jimah, Ph.D.

Guest Editor

Journal Requirements:

2. Please provide separate figure files in .tif or .eps format.

Reviewers' comments:

Reviewer's Responses to Questions

**Comments to the Author**

1. Does this manuscript meet PLOS Global Public Health’s publication criteria?

Reviewer #1: Yes

Reviewer #2: Yes

2. Has the statistical analysis been performed appropriately and rigorously?

Reviewer #1: Yes

Reviewer #2: Yes

3. Have the authors made all data underlying the findings in their manuscript fully available (please refer to the Data Availability Statement at the start of the manuscript PDF file)?

Reviewer #1: Yes

Reviewer #2: Yes

4. Is the manuscript presented in an intelligible fashion and written in standard English?

Reviewer #1: Yes

Reviewer #2: Yes

Reviewer #1: Thank you for the opportunity to review this important and well-constructed manuscript. Below are detailed comments and suggestions for improvement.

Abstract:

-The title is clear and compelling. You might consider making it slightly more active, e.g., “Who Gets Missed in Zambia’s Antenatal HIV Testing? Persistent Inequities Despite Routine Screening” to strengthen impact.

-Consider adding the sample size and clarifying that HIV testing was self-reported. This would enhance the completeness and transparency of the summary.

-The conclusion is strong. If possible, include a brief mention of specific strategies, e.g., mobile testing or peer education that were effective or promising based on your findings or the literature.

Introduction

-The rationale is well-articulated, and the introduction flows logically from global context to local policy gaps.

-When introducing the topic of inequities, it would strengthen the reader’s understanding to briefly name a few mechanisms (e.g., structural barriers, stigma, male partner dynamics) that may underlie disparities in testing.

Methods

-The study design is appropriate, and inclusion criteria are clearly defined. Harmonization of DHS variables across years is a strength.

-Please clarify whether the HIV testing outcome includes both receipt of the test and receipt of the result, or only one of these. That affects interpretation of self-reported testing uptake.

-The use of survey-adjusted logistic regression models with appropriate DHS weighting (v005, v021, v022) is appropriate and adds credibility to national-level estimates.

-The application of a quasi-binomial error structure is a reasonable choice to account for overdispersion. That said, briefly explaining why this approach was selected would benefit readers unfamiliar with this modeling nuance.

-You could consider reporting diagnostics or model performance measures (e.g., pseudo R², ROC curve, or Hosmer-Lemeshow test) to further demonstrate the robustness of your regression models.

-Given the importance of education and wealth as predictors, a check for multicollinearity (e.g., VIF) would also help support the independence of their effects. If not already performed, including a sensitivity analysis or model stability check would be a useful addition.

Results

-The descriptive statistics and regression findings are clearly presented.

-In Table 1, you may consider adding a row that summarizes HIV testing rates per year to allow quick reference.

-In the text, the observation that testing increased from 87% to 95% is an important achievement that’s well-described. The narrative on improvements in education could be deepened with a note about how this may partially explain increased testing, particularly in rural areas in the discussion.

-The stratified regression analysis is a highlight. It reveals nuanced patterns that would be lost in pooled models. Presenting these separately by rural and urban populations is a thoughtful decision and adds to the policy relevance of the findings.

-One extension that may be worth considering is testing interaction terms between key predictors and place of residence (e.g., education × rural vs urban) to statistically confirm whether disparities differ significantly by geography.

-A visual representation (e.g., forest plot of adjusted odds ratios grouped by domain) might improve accessibility and visual impact for readers.

Discussion

-The discussion effectively summarizes the findings and relates them to broader regional trends.

-The education and wealth gradients are clearly explained. The rural–urban stratification is particularly valuable, and I appreciated your observation that socioeconomic predictors are stronger in rural areas.

-You highlight that adolescents are less likely to be tested, especially in rural settings. This is an important point. You could suggest school-based HIV education or adolescent ANC peer navigators as promising strategies—especially given that youth-focused HIV services are still limited in many settings.

-The mobile clinic and peer-led education recommendations are appropriate. Consider adding a comment about ensuring reliable supply chains and infrastructure in rural clinics, which can affect trust and follow-up in antenatal HIV care.

Limitations and Strengths

-The limitations are appropriate and honestly presented. It might be worth noting that social desirability bias in self-reporting could differ by setting, and may actually lead to underestimation of disparities.

-One of the key strengths is your focus on women already engaged in ANC. That this group still experiences such disparities makes your findings even more compelling for policymakers. Emphasizing this in the discussion may help frame your policy recommendations as “last-mile” solutions for already-engaged populations.

Minor Language Suggestions

-Be consistent with abbreviations: once “antenatal care (ANC)” is introduced, use ANC throughout.

-Avoid repetitive phrasing like “less educated, poorer, and rural-dwelling women” as this appears several times. Rephrasing or grouping these differently would improve the narrative flow.

-Instead of “engaged with the health system,” you might say “already accessing ANC services” for clarity.

Reviewer #2: Very nice use of multiple DHS surveys to examine trends and inequities in HIV testing in antenatal clinics in Zambia. The manuscript is well-written, and research questions and secondary data analysis methods are described clearly with sufficient detail. The rural/urban stratified analysis is important and a strength of the paper. There is some redundancy between tables and figures, but still helpful to see both. Overall, the manuscript shows that despite various programs and interventions in Zambia, there are persistent gaps in ANC HIV testing, particularly in rural areas, and among the least educated and poorest women.

**Do you want your identity to be public for this peer review?** For information about this choice, including consent withdrawal, please see our Privacy Policy

Reviewer #1: No

Reviewer #2: No

---

## [Editor Report · Decision Letter 1]

10 Nov 2025

Who Gets Missed in Zambia’s Antenental HIV Testing? Persistent Inequities Despite Routine Screening Policies in Zambia.

PGPH-D-25-01877R1

Dear Dr Nyirenda,

We are pleased to inform you that your manuscript 'Who Gets Missed in Zambia’s Antenental HIV Testing? Persistent Inequities Despite Routine Screening Policies in Zambia.' has been provisionally accepted for publication in PLOS Global Public Health.

Best regards,

Tamara Jimah, Ph.D.

Guest Editor
